# Robust Image Segmentation Quality Assessment

**Leixin Zhou**                                                    leixin-zhou@uiowa.edu
**Wenxiang Deng**                                              wenxiang-deng@uiowa.edu
**Xiaodong Wu**                                                xiaodong-wu@uiowa.edu
*Department of Electrical and Computer Engineering, University of Iowa, USA*

## Abstract

Deep learning based image segmentation methods have achieved great success, even having human-level accuracy in some applications. However, due to the black box nature of deep learning, the best method may fail in some situations. Thus predicting segmentation quality without ground truth would be very crucial especially in clinical practice. Recently, people proposed to train neural networks to estimate the quality score by regression. Although it can achieve promising prediction accuracy, the network suffers robustness problem, e.g. it is vulnerable to adversarial attacks. In this paper, we propose to alleviate this problem by utilizing the difference between the input image and the reconstructed image, which is conditioned on the segmentation to be assessed, to lower the chance to overfit to the undesired image features from the original input image, and thus to increase the robustness. Results on ACDC17 dataset demonstrated our method is promising.

**Keywords:** Robust, Segmentation Quality, Deep Learning

## 1. Introduction

Segmentation quality assessment with the absence of ground truth, which estimates segmentation accuracy without human or expert intervention, is of high interest in medical imaging research and clinical fields. In many applications, the deep learning based segmentation methods can even achieve expert-level accuracy. However, in practice, deep learning methods may fail due to many factors: such as domain shift (Patel et al., 2015), adversarial noise, and low image quality. Therefore predicting segmentation quality without ground truth would be very crucial and of high interest for the downstream analysis.

One straightforward idea is to predict segmentation quality using a CNN regression network, where the image and its segmentation are concatenated as different channels to feed into the network (Robinson et al., 2018b,a). However, that state-of-the-art method suffers the robustness problem if the input images have a different distribution from that of those training datasets for the regress network. This can be demonstrated with adversarial attacks, in which it involves adding hand-crafted perturbations to the images drew from the distribution of training data and leading to misbehave for deep neural networks.

Inspired by the work of representation learning and factorization (Mirza and Osindero, 2014; Chartsias et al., 2018), we propose to improve the prediction robustness by extracting features directly related to the segmentation. More precisely, we propose to utilize the difference of the original input image and the reconstructed image conditioned on the input image and the input segmentation. Our work is most related to Kohlberger *et al.*'s work (Kohlberger et al., 2012), in which the quality assessment score is estimated by re-

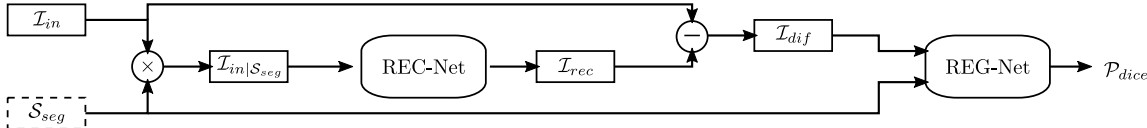

Figure 1: The work flow of proposed segmentation quality assessment method.

gression based on numerous statistical and energy measures from segmentation algorithms. Our method also shares merits of unsupervised lesion or outlier detection (Schlegl et al., 2017; Baur et al., 2018; Chen and Konukoglu, 2018; Pawlowski et al., 2018; Seeböck et al., 2016; Alaverdyan et al., 2018), where only normal data (ground truth segmentation in our scenario) is utilized in the training of the reconstruction network.

**Contributions**: In this paper, we propose to make use of features directly related to segmentation to improve the robustness of the quality regression network for segmentation quality assessment. To achieve this goal, we have developed two CNNs: one is a reconstruction network (REC-Net), which aims to reconstruct the original image from the image masked by the provided segmentation; the other is a quality regression network (REG-Net), which predicts the segmentation quality based on the reconstruction difference image and the provided segmentation. Our experiments on ACDC17 dataset [1] have demonstrated highly promising performance of the proposed method.

## 2. Method

Assume the input image, its ground truth segmentation and the candidate segmentation (to be assessed) are $\mathcal{I}_{in} \in \mathbb{R}^{n \times n}$, $\mathcal{S}_{gt} \in \mathbb{Z}^{n \times n}$ and $\mathcal{S}_{seg} \in \mathbb{Z}^{n \times n}$, respectively. It is trivial to apply any metric functions (e.g. Dice, Jaccard scores) to the pair $\mathcal{S}_{gt}$ and $\mathcal{S}_{seg}$ to get the ground truth segmentation quality score, e.g. $\mathcal{GT}_{dice}$. However, the absence of $\mathcal{S}_{gt}$ makes generating Dice prediction $\mathcal{P}_{dice}$ non-trivial.

The flow of the proposed method is demonstrated as in Fig. 1. We use $\mathcal{I}_{in|\mathcal{S}}$ to represent the image with the segmented target $\mathcal{S}$ being masked by zero, in which $\mathcal{S}_{ij} = 1$ if the corresponding pixel belongs to the target; otherwise $\mathcal{S}_{ij} = 0$. More specifically, $\mathcal{I}_{in|\mathcal{S}} = \mathcal{I}_{in} \cdot (1 - \mathcal{S})$. In other words, all pixels that are labeled by $\mathcal{S}$ as the target object in $\mathcal{I}_{in}$ are set to zero intensity. The reconstructed image using the proposed reconstruction network (REC-Net) from $\mathcal{I}_{in|\mathcal{S}}$, is denoted as $\mathcal{I}_{rec}$. The difference image $\mathcal{I}_{dif}$, which serves as one input channel to the quality regression network (REG-Net), is defined as: $\mathcal{I}_{dif} = \mathcal{I}_{in} - \mathcal{I}_{rec}$. The output of REG-Net $\mathcal{P}_{dice}$ is the predicted score for the segmentation quality.

During the training of REC-Net, *only* pairs of $\mathcal{I}_{in}$ and its $\mathcal{S}_{gt}$ are utilized. The rationale behind is that the REC-Net is trained to well recover the original input image only when $\mathcal{S}$ is a *good* segmentation. However, during the training of REG-Net, segmentations of different quality have to be used to teach the REG-Net the quality measure. The REC-Net and the REG-Net have a U-net and Alex-net architectures, respectively.

---

1. https://www.creatis.insa-lyon.fr/Challenge/acdc/databases.html

| Method | $\epsilon = 0$ | $\epsilon = 0.05$ | $\epsilon = 0.1$ | $\epsilon = 0.2$ | $\epsilon = 0.3$ |
|---|---|---|---|---|---|
| Robinson *et al.* | 0.04±0.05 | 0.08±0.06 | 0.11±0.07 | 0.14±0.08 | 0.16±0.09 |
| proposed | 0.04±0.05 | 0.07±0.06 | 0.09±0.06 | 0.09±0.07 | 0.12±0.09 |

Table 1: Mean absolute errors of dice prediction under different levels of adversarial attack.

## 3. Experiments

### 3.1. Data

To validate the proposed segmentation quality assessment method, we utilize a public dataset: Automated Cardiac Diagnosis Challenge (ACDC) MICCAI challenge 2017. For our experiments, only segmentation of left-ventricular myocardium (LVM), which is very challenging, was considered. To train the REG-Net, segmentations of different quality have to be generated first. In our experiments, in contrast to random forests used in (Robinson et al., 2018a), U-nets with different depths, different number of starting filters and different training epochs, were applied to generate the simulated segmentations with different quality.

### 3.2. Adversarial attacks generation

We compared the robustness of our proposed method with respect to adversarial attacks against the state-of-the-art methods (Robinson et al., 2018a, 2017, 2018b). We applied a simple fast gradient sign method (Kurakin et al., 2016) to generate the adversarial images for REG-Net to conduct our experiments. Only adversarial attacks on the original images $\mathcal{I}_{in}$ and the difference image $\mathcal{I}_{dif}$ were considered and no changes were made to $\mathcal{S}_{seg}$.

### 3.3. Performance comparison

The mean absolution error (MAE) of the Dice scores, MAE $= \frac{\sum_{i=1}^{SN} |\mathcal{P}_{dice}^i - \mathcal{GT}_{dice}^i|}{SN}$, was utilized as the metric, where $SN$ is total number of slices in the test set. The results without adversarial attacks are shown in column $\epsilon = 0$ in Table. 1. As can be seen, when there is no attack, the proposed method works as well as Robinson *et al.*'s (Robinson et al., 2018a, 2017, 2018b). The performance when having attacks is demonstrated in the right most four columns in Table. 1. It can be noticed that for both methods, the MAEs are monotonically non-decreasing as the attack level increases. However, the proposed method has a smaller increasing rate and works better than Robinson *et al.*'s.

## 4. Conclusion

In this paper, a robust method for segmentation quality assessment has been proposed. We make use of the image difference between the input image and the reconstructed image using our proposed image reconstruction network (REC-Net), as the feature image for the quality score regression network (REG-Net). Results on ACDC17 dataset verified our method is more robust.

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
