# OpenReview forum: "Robust Image Segmentation Quality Assessment "
_MIDL.io/2020/Conference — MIDL 2020_

### Official Review · AnonReviewer1 · 2020-02-20
**some potential but results are not convincing; Contribution is over-stated.**

**Rating:** 2
**Confidence:** 4

**Review:**

This work outlines a method to estimate segmentation quality without learning
from ground truth. Quality assurance is an important topic and should feature
at MIDL. However, this abstract is very casually written ('people proposed',
'the network suffers', etc.) and some parts of the motivation are too broad,
e.g. how are robustness and adversarial attacks connected? The first thing that
would come to my mind regarding robustness is rather domain shift as stated
in the Introduction, and probably the last are adversarial attacks.
However, the abstract improves later on and reasonable justifications are given.
The authors motivate their contribution de-emphasising irrelevant/adversarial features
which seems reasonable.

The presented results are not convincing. The differences To Robinson et al.
seem to be minimal and well within the confidence margins. It is questionable
if adding a task-dissimilar self-supervision loss should be able to improve an approach
like Robinson et al. significantly at all. I think this needs thorough discussion
and the abstract should avoid overstating that their method improves robustness
of automated segmentation quality assessment.

---

### Official Review · AnonReviewer2 · 2020-02-25
**improved segmentation performance by using QA**

**Rating:** 4
**Confidence:** 4

**Review:**

The quality and clarity are high in this work.
Pros:
The method is designed for unsupervised lesion or outlier detection.
Only normal data is utilized in the training of the reconstruction network.
The method achieves better mean absolute errors of dice prediction under different levels of adversarial attack.

Cons:
The performance is evaluated on the adversarial attacks. The performance on lesion and outlier is not evaluated.

---

### Official Review · AnonReviewer4 · 2020-03-11
**Interesting idea and evaluation against adversarial examples**

**Rating:** 3
**Confidence:** 4

**Review:**

The authors proposed a method for segmentation quality assessment. The method consists of learning a reconstruction network for the masked input images and a regression network for quality assessment. The reconstruction network aims to only faithfully reconstruct input images masked correctly by the segmentation, while the regression network learns to assess the quality by looking at segmentation of different quality. The robustness of the proposed method is supported by quantitative evaluation with comparison to a baseline method.

The underlying idea interestingly links to the earlier works in unsupervised detection and couples it with quality assessment. The paper is well structured with clear contribution and provides the key results. The results show that the proposed method is more robust against adversarial attacks.

Additionally, a few concerns are:

1) REG-Net is trained to assess the segmentation quality by providing it with images of different segmentations, . What is the metric to assess such segmentations, or in other words, what's the ground-truth for $P_{dice}$, and how is it obtained?

2) U-net is used as the framework for REC-Net. U-net have skip connections to preserve details for segmentation, however, using skip-connections for the very first layers for reconstruction may leak much information and make the task very easy, is this the case for REC-Net in this work?

3) It may be useful if the authors can give the loss function for the proposed method in the paper.

---

### Official Review · AnonReviewer3 · 2020-03-12
**Interesting idea but rough in execution**

**Rating:** 2
**Confidence:** 5

**Review:**

This work tries to address robustness in image segmentation quality assessment by training 2 networks. A reconstruction network and a regression network. The result seems fine but there are some problems to attend.

1. The comparison is not clear. The authors do not specify which of the method they compared in the reference, where at least 2 methods were applied.
2. The $\epsilon$ representing the adversarial attack level is numerically different from the referenced criterion. The authors need to address why there is a numerical difference and explain why they specifically choose these levels of adversarial attack.
3. The claim ‘Our method also shares the merits of unsupervised lesion or outlier detection’ is not convincing. These referenced methods merits are not just training on normal data but more because of training on normal data allowing them modeling the distribution implicitly or explicitly using these generative models, whereas in this paper no such modeling on normal data is shown.
4. This work overstates ‘developed two CNNs’, where it just takes advantage of a U-Net and an Alex-net. The authors need to tone down on this claim or perform major modification on the architecture or objectives.

---

### Meta-Review · Area_Chair1 · 2020-04-06
**MetaReview of Paper93 by AreaChair1**

**Rating:** 3

**Metareview:**

All the authors agree that the method has some potentials and the idea is interesting. I think that it would be interesting to be included as a short paper in MIDL.

**Paper Type:**

both

---

### Decision · Program_Chairs · 2020-04-11

Accept